# In-Water Fish Body-Length Measurement System Based on Stereo Vision

**DOI:** 10.3390/s23146325

**Published:** 2023-07-12

**Authors:** Minggang Zhou, Pingfeng Shen, Hao Zhu, Yang Shen

**Affiliations:** Agricultural Machinery Engineering Research and Design Institute, Hubei University of Technology, Wuhan 430068, China; 19911070@hbut.edu.cn (M.Z.); shenpingfeng@hbut.edu.cn (P.S.); zhuhao@hbut.edu.cn (H.Z.)

**Keywords:** binocular vision, image processing, fish body-length measurement, grab cut segmentation, water refraction, aquaculture

## Abstract

Fish body length is an essential monitoring parameter in aquaculture engineering. However, traditional manual measurement methods have been found to be inefficient and harmful to fish. To overcome these shortcomings, this paper proposes a non-contact measurement method that utilizes binocular stereo vision to accurately measure the body length of fish underwater. Binocular cameras capture RGB and depth images to acquire the RGB-D data of the fish, and then the RGB images are selectively segmented using the contrast-adaptive Grab Cut algorithm. To determine the state of the fish, a skeleton extraction algorithm is employed to handle fish with curved bodies. The errors caused by the refraction of water are then analyzed and corrected. Finally, the best measurement points from the RGB image are extracted and converted into 3D spatial coordinates to calculate the length of the fish, for which measurement software was developed. The experimental results indicate that the mean relative percentage error for fish-length measurement is 0.9%. This paper presents a method that meets the accuracy requirements for measurement in aquaculture while also being convenient for implementation and application.

## 1. Introduction

Length, as the most basic piece of information on fish, is an essential monitoring parameter in aquaculture engineering. During the breeding process, to create a favorable growing environment, fish must be reared separately according to size [1,2,3]. To meet market requirements, fish should be graded by size when they mature to increase value [4,5]. In addition, determining the size of the fish in a breeding pond can help monitor growth and even predict sex and age [6,7]. Fish quality is another piece of basic information. A relationship exists between fish length and quality, and fish quality can be determined by obtaining fish length [8,9,10,11,12]. However, the length of a fish is typically measured manually with a ruler, which can cause damage or even lead to the death of fish. The accuracy of the results is also easily affected by subjective human factors. As a fast, accurate, non-destructive, and economical measurement method, machine vision systems have been applied to fish body-length measurement [13,14,15], overcoming the above-mentioned disadvantages. For fish in water, it is more difficult to obtain fish size information without contact because the fish are swimming.

With the rapid development of machine vision and image processing technologies, several methods are available for measuring the size of fish in water. There are two major ways to measure the size of fish in water using vision technology: putting a camera device into the water or placing cameras around the tank. Torisawa et al. [16] used a digital stereo camera system to monitor bluefin tuna swimming freely in a net cage in three dimensions. A direct linear transformation (DLT) method was adopted to estimate the fork length of tuna, with an error ratio of < 5%. Shafait et al. [17] presented a semi-automatic method for capturing underwater stereo–video measurements to estimate fish length. The results showed that the error of the fork length measurement was less than 1% of the true length, reducing the operation time by at least six times compared to manual measurement. Subsequently, a non-invasive, fully automated procedure was proposed to obtain highly accurate length estimates of bluefin tuna, with a 3% error in fish-length estimation in up to 90% of the samples [18]. Tatsuya et al. [19] used multiple stereo images to monitor the growth of free-swimming red snapper cultured in a net tank and determined the effectiveness of stereo image measurements for monitoring the growth process of fish.

Meanwhile, Shi [20] presented a stereo vision system with a procedure that obtains the *oplegnathus punctatus* fish body length based on LabVIEW, with a mean relative error of <2.55%. Savinov et al. [21] used two video cameras for data acquisition: one located above the aquarium and one on the long side; rainbow trout length was measured with an error of 5%. Similarly, front and side images of zebrafish in a seed tank were captured continuously using a dual synchronous orthogonal webcam to identify the outline and position, and the length was then measured with an average estimation error of approximately 1% [22]. Abe et al. [23] presented an automated method for measuring the length of free-swimming fry fish in a small aquarium; the standard deviation of the error rate was less than 5%.

According to the above studies, when the camera was placed in the water for measurement, it was difficult to adjust the position, and the underwater environment was more complex. Image acquisition quality was affected by fish body obfuscation, bending, and imaging angle. The measurement of fish in aquariums was easier with a camera, but the measurement was affected by the bending state of the fish and refraction by the water body and glass. However, these methods are difficult to apply to factory farming environments.

This study mainly focused on binocular stereo vision technology, collecting the top view image of underwater fish using a handheld or fixed binocular camera on the water and then obtaining the fish body length using an image processing algorithm.

## 2. Materials and Methods

This paper proposes an adjustable distance measurement method based on the RealSense D435 binocular camera to accurately and quickly measure the size of fish in water. The procedure of this study included four main steps: image acquisition, image segmentation, consideration of water refraction, and extraction of measurement points and calculation. These steps are important for accurately measuring and analyzing the data. Each step involves specific techniques and processes that are crucial to the overall success. The overall structure of the method proposed in this paper is shown in Figure 1.

In this study, RGB and depth images were acquired using the RealSense D435 binocular stereo camera. The acquisition distance of the camera could be adjusted arbitrarily, making it more convenient for practical measurements.This study utilized a high-precision interactive segmentation algorithm that accurately segments the target fish based on user requirements. This algorithm is effective in handling complex segmentation scenarios, resulting in refined segmentation outcomes. If the segmentation result is unsatisfactory, it can be segmented again.This paper presents an algorithm for determining the curvature of fish, enabling differentiation between curved and straight fish. This algorithm employs various measurement key point extraction methods for different types of fish to ensure accuracy. The measurement key points are projected into 3D space to calculate the 3D spatial coordinates of the measurement points. The distance between two points is then obtained from the 3D coordinates, resulting in a more precise measurement method.This study analyzed the effect of water refraction on the measurement and aimed to effectively reduce it.

### 2.1. Experimental Setup

#### 2.1.1. Experimental Platform

To ensure the credibility of our experiment, we developed our own experimental measurement platform. The platform consists of several components, including a breeding bucket, oxygenation pump, overhead tripod, upper computer, lighting source, and RealSense D435 camera, as shown in Figure 2. The upper opening diameter of the breeding barrel was 62 cm, the lower opening diameter was 46 cm, the height of the barrel was 71 cm, and the water depth in the barrel was 50 cm. The distance between the camera and water surface could be adjusted at will. The RealSense D435 sensor is an affordable and compact device that operates without the need for an external power source. It comprises two infrared cameras, an infrared dot matrix projector, and an RGB camera. The infrared cameras and projector work together to produce depth images, while the RGB camera captures RGB images. The camera measures 90 mm × 25 mm × 25 mm and has a depth field of view of 87 × 58. The RGB sensor has a field of view of 69 × 42. The effective measurement range is between 0.2 and 10 m.

The Processing Host used for this project was equipped with an Intel^®^ Core™ i5-7300HQ CPU @ 2.50 GHz, 64-bit Windows 10 operating system, and 24 GB of system memory. The development was performed using Microsoft Visual Studio 2019, and the measurement algorithm was developed in C++. The depth camera series utilized the RealSense 2.0 software development kit, which facilitated operations such as turning the camera on and off and aligning depth with RGB data. Image processing operations, such as saving, displaying, and processing, were accomplished using the OpenCV computer vision library.

The subjects in this experiment were 10 perch that differed in body size and color, with a body length distribution of 250–320 mm.

#### 2.1.2. Image Acquisition

Owing to the different coordinate systems of the RealSense D435i’s left and right depth cameras and RGB color camera, there was a misalignment between the acquired color and depth images. Thus, it was necessary to correspond the pixel points in the depth image to those in the RGB image. The alignment process involves converting the 2D points in the depth image to their corresponding 3D points in the world coordinate system. These 3D points are then projected onto the color image for a more accurate representation. The camera resolution was 640 × 480 pixels, which meets the necessary requirements.

### 2.2. Image Segmentation

To accurately determine the size of a fish, it is necessary to perform image segmentation to eliminate any background noise and identify the precise location of the fish within the image coordinate system. The selective segmentation of the entire fish is required when multiple fish are present in the acquired RGB images. The Grab Cut algorithm [24,25,26,27] is an interactive segmentation algorithm based on graph theory, with fewer interactive operations and higher segmentation accuracy. In recent years, this algorithm has been used by many researchers to achieve impressive results in image and video segmentation. The Grab Cut algorithm requires the foreground and background to be selected to create a Gaussian mixture model (GMM) for the foreground and background regions. The parameters of the GMM are then iteratively adjusted to obtain the minimum Gibbs energy for the assigned pixels, defined as
(1)E(α,k,θ,z)=U(α,k,θ,z)+V(α,z)
where U is the regional term indicating the sum of penalties for all pixels classified as background or target; *α* is a differentiating vector for foreground and background, *α* ∈ [0, 1]; *k* is the model component of the GMM; *θ* represents the corresponding GMM parameters; *z* is the image grayscale array; and *V* is the boundary term.
(2)$$U(\alpha ,k,\theta ,z)=\sum_{n}D(\alpha_{n},k_{n},\theta ,z_{n})$$
(3)$$D(\alpha_{n},k_{n},\theta ,z_{n})=-\log P(Z_{n}|\alpha_{n},k_{n},\theta_{n})-\log\pi (\alpha_{n},k_{n})$$
where *P* (•) is the Gaussian probability distribution, and *π* (•) is the mixture weighting factor.
(4)$$V(\alpha ,z)=\gamma \sum_{(m,n)\in C}[\alpha_{n}\neq \alpha_{m}]\exp-\beta ||z_{m}-z_{n}|{|}^{2}$$
where *γ* is the weight factor of the boundary term, and $\beta =(2{\langle (z_{m}-z_{n})}^{2}\rangle {)}^{-1}$.

Next, we identify the region of interest (ROI) by manually labeling it with a rectangle. This ROI is set as the foreground, while the rest of the image is considered the background. The Grab Cut algorithm can obtain the foreground image and thus achieve segmentation. However, owing to the influence of lighting and the environment, or when the selected foreground contained more background, a mistake would occur, splitting the background into the foreground. An algorithm for contrast adaptive adjustment was employed for improvement. The RGB image is copied and converted into a gray image with only one channel. The contrast between the fish and background is improved by using a linear stretch transformation algorithm. Increasing the overall brightness of the image eliminates the effects of lighting. The input image for the Grab Cut algorithm was a three-channel image. Each pixel value in the R, G, and B channels of the original RGB image is set as the pixel value of the single-channel image. The input image processed by the Grab Cut algorithm is shown in Figure 3b, and the binarized output results are shown in Figure 3c,d. The image segmentation flow in the work is shown in Figure 4.

### 2.3. Measurement Method

After segmentation, minor noise may be present in the image. To address this, we recommend utilizing erosion and expansion image processing operations. Specifically, we suggest first eroding the image and then expanding it to a kernel size of 3 × 3. To distinguish between the curved and straight states of the fish in a segmented image, an algorithm must be developed. First, a contour finding algorithm can be employed to locate the outline of the fish body. Once the outline is found, we select it with a minimum bounding rectangle. The four vertices of the rectangle are returned, with the two short sides connecting the rectangle denoted as *ab* and *cd*. The intersection of the line *ab* with the contour is denoted as *e*(*ue*,*ve*), while the intersection of the line *cd* with the contour is denoted as *f*(*uf*,*vf*). As shown in Figure 5b, the midpoint of the smallest outer rectangle of the contour is point *o*, connecting the lines *eo*, *of*, and *ef*. In ∆*eof*, the value of ∠*eof* can be solved according to the law of cosines, and then the supplementary angle of ∠*eof* is calculated as *θ*. The parameter *θ* is utilized to determine if the fish is in a bent state or not. In Figure 5b, where *θ* = 16.9°, it is noticeable that the fish appears curved. However, in Figure 5a, where *θ* = 1.25°, the straight line *ef* is approximated as a straight line that passes through the midpoint of the rectangle, and the fish appears to be straight. By establishing a threshold value of *θ* = 5°, it is possible to determine if the fish is bent or not.

#### 2.3.1. Linear Measurement

When judging that the fish is straight, as in Figure 5a, points *e* and *f* are the best measurement points. The depth data of points *e* and *f* are read from the depth map and corrected using the refraction model described in the following section. By converting the coordinates *e*(*u_e_*,*v_e_*) and *f*(*u_f_*,*v_f_*) into three-dimensional coordinates, $E(X_{E},Y_{E},Z_{E})$ and $F(X_{F},Y_{F},Z_{F})$, the actual body length of the fish can be precisely calculated using the following Euler distance equation:(5)$$L=\sqrt{{\left(X_{E}-X_{F}\right)}^{2}+{\left(Y_{E}-Y_{F}\right)}^{2}+{\left(Z_{E}-Z_{F}\right)}^{2}}$$

#### 2.3.2. Bend Measurement

For curved fish, a skeleton extraction method [28,29,30,31] was applied to extract the mid-axis of the fish, and the results are demonstrated. In this study, we used the Zhang-Suen refinement algorithm [32] for skeleton extraction. This algorithm was chosen due to its fast computational speed and ability to maintain curve connectivity. The algorithm iteratively marks the pixels that are not skeleton points, loops through all pixel points, and removes the pixels that satisfy certain conditions until no pixels satisfy the conditions in a certain iteration. However, the direct use of the algorithm in this study may have resulted in the appearance of small branches, which can be defined as burrs and should be removed. To eliminate this, we proceeded as follows.

The number of fields around a pixel is defined as *N* = 1, 2, …, 8. The pixel points are defined as follows:(6)$$\left\{\begin{array}{l} N=1\;\mathrm{endpoints}\\ N=2\;\mathrm{skeleton}\;\mathrm{points}\\ N\geq 3\;\mathrm{nodes} \end{array}\right.$$

Step 1. The optimization of the Zhang-Suen algorithm involves removing the corner points displayed in Figure 6. The corner points are defined as protrusion points in four directions. This action reduces the number of nodes without altering the connectivity of the skeleton.

According to Figure 7, a corner point can be formed in a certain direction when pixel points P, P2, and P8 each have a value of 1. However, this is only a necessary but not sufficient condition for the formation of corner points. Furthermore, when pixel P is a corner point, the pixel points P4, P5, and P6 must simultaneously be zero. By observing Figure 6, it can be determined that a pixel point is confirmed as a corner point when it meets one of the following conditions:(7)$$\left\{\begin{array}{l} \left(P2=P8=1)\&(P4=P5=P6=0\right)\\ \left(P2=P4=1)\&(P6=P7=P8=0\right)\\ \left(P4=P6=1)\&(P1=P2=P8=0\right)\\ \left(P6=P8=1)\&(P2=P3=P4=0\right) \end{array}\right.$$

Step 2. The number of endpoints of the entire image is judged, and if it is greater than 2, burr processing is performed.

Step 3. To determine the burr branch, endpoints with the longest distance between any two points among all endpoints are selected, and the line between these two points is the skeleton line, while the branch where the remaining endpoints are located is the burr.

Step 4. To iteratively deburr branches, delete the endpoint, find the pixel point in the eight fields of the endpoint, and determine the numbers of the eight fields at that point. If N is equal to 1, it becomes the endpoint, and the above operation is repeated until N is greater than 2 or until the number of deletions reaches the set threshold and then stops.

Step 5. Determine whether there will be a break, and if so, connect the breakpoints to ensure that there are only two endpoints.

The results of deburring are shown in Figure 8, and the deburring flowchart is shown in Figure 9.

In Figure 10a, because the median line obtained after the Zhang-Suen algorithm processing has parts missing at the ends, we find the two endpoints *g* and *h* of the curve. The points where the contour and smallest outer rectangle intersect are *e* and *f,* connecting the lines *eg* and *hf*; then, the curve *ef* is the length of the fish in the image. To solve for the length of the curve *ef,* we extend *ef* to the point *m* such that the length of the line *em* is equal to the length of the curve *ef*, as shown in Figure 10b. Finally, *m*(*u_m_*,*v_m_*) is taken as the measurement point of the tail, and *e*(*u_e_*,*v_e_*) is taken as the measurement point of the head. Then, the depth data taken from the original *f* point are assigned to the *m* point, which completes the “straightening” of the fish body, and the body length of the fish when it was bent is obtained after refraction correction.

### 2.4. Correction of Water Refraction

Regardless of the camera placement, whether submerged in water, positioned above the water surface, or surrounding the water tank, the light entering the camera undergoes refraction, thereby affecting the measurement to some extent. When light propagates through water and air [33,34,35], refraction occurs at the water-air interface, as shown in Figure 11.

Point $P(X_{w},Y_{w},Z_{w})$ is a point in three-dimensional space, $Q_{a}(x_{a},y_{a})$ is a point on the imaging plane without the refraction of light from water, and $Q_{w}(x_{w},y_{w})$ is a point on the imaging plane after the refraction of light from water. Figure 11 demonstrates that the imaging object will appear larger after refraction by water. It is crucial to establish a correlation between the measurement of an object in air and refracted image size. There exists a certain relationship between the two points, $Q_{a}=KQ_{w}$, which can be known according to the geometric relation.
(8)$$K=\frac{Q_{a}}{Q_{w}}=\frac{h}{Z}+\left(1-\frac{h}{Z}\right)\frac{\tan\beta }{\tan\alpha }$$
where *h* is the distance from the camera to the water surface, which can subtract the water depth in the barrel from the placement distance of the camera. The variable *Z* represents the distance from the object point P in the water as read by the camera. *α* is the angle of refraction in air after the light is refracted by water, *β* is the angle of incidence of light in water.

According to the law of refraction,
(9)$$n_{a}\sin\alpha =n_{w}\sin\beta$$
where $n_{a}$ is the refractive coefficient of light in air ($n_{a}=1$), and $n_{w}$ is the refractive coefficient of light in water ($n_{w}=1.333$).

The value of α can be acquired from the obtained imaging plane:(10)$$\sin\alpha =\sqrt{\frac{x^{2}{}_{w}+y_{w}{}^{2}}{x^{2}{}_{w}+y_{w}{}^{2}+f^{2}}}$$

From the pixel and image coordinate systems, it can be concluded that
(11)$$\left\{\begin{array}{l} u=\frac{x}{dx}+u_{0}\\ v=\frac{y}{dy}+v_{0} \end{array}\right.$$
where $u_{0}$ and $v_{0}$ are the coordinates of the main points of the RGB camera. This converts the measured points (*u*, *v*) from the acquired RGB image into imaging points (*x*, *y*) in the imaging plane.

Therefore, the relationship between the refracted and un-refracted imaging points of light in the imaging plane can be obtained from the pixel positions in the acquired RGB images.

As shown in Figure 12, *Pa* is the distance measured in air, and *Pw* is the distance measured after refraction. According to the triangle similarity theorem, we can obtain the distance of point *P*.
(12)$$Z=\frac{B\cdot f}{d}$$

We found that the distance measured after the light is refracted by water will be smaller than that in air. In fact, the distance of the object read out using the camera is smaller than the actual distance measured, thus requiring a correction of the depth distance in the first place. For the left camera $x_{l}\prime =Kx_{l}$, the right camera has $x_{r}\prime =Kx_{r}$, according to the parallax $d=K(x_{l}-x_{r})$, and the corrected distance is $Z\prime =\frac{B\cdot f}{Kd}$.

Then, the corrected distance was applied to correct the error generated by the refraction of the image and to find *K_w_*. The relationship between the camera and pixel coordinate systems was obtained as the following equation:(13)$$\left\{\begin{array}{l} X_{w}=\frac{K_{w}(u-u_{0})}{f}\times Z_{w}\\ Y_{w}=\frac{K_{w}(v-v_{0})}{f}\times Z_{w}\\ Z_{w}=Z\prime \end{array}\right.$$

Thus, the 2D coordinates in the image coordinate system can be transformed into 3D coordinates in the camera coordinate system, and the distance between two points on the image in real space can be calculated using the spatial distance formula.

### 2.5. Perch Length Measurement Software

An MFC-based interface program was designed to measure the body length of perch. The program comprises four main components: data input, video presentation, image processing, and result display. The interface of the software is shown in Figure 13.

**Data input.** To accurately account for water refraction, this module requires the input of the fixed camera height and water depth in the breeding bucket. If these values are not entered, the obtained results will not take into account the effect of water refraction.**Video presentation.** The demo display section allows the user to switch the video on and off and obtain the RGB and depth data streams. Upon clicking the Save Image button, the RGB image is saved and displayed in the image processing area, while the depth image is saved in the backend.**Image processing.** This part mainly achieves the above image segmentation function. The contrast of the image is adjusted, and the Region of Interest (ROI) is selected by drawing a rectangle around the fish. Next, the image segmentation button is clicked to obtain the segmentation result. If the segmentation result is not satisfactory, the original image can be restored and the previous steps repeated. Once satisfied with the segmentation result, the Calculate button is clicked to determine the length of the fish, which will be displayed in the result area.**Result display.** This mainly shows the coordinates and depths of the measurement points, bend, and length of the target fish.

## 3. Results

To validate the accuracy of the proposed contactless method for measuring the body length of perch in water, as well as the feasibility of adaptive measurement and precision of calculation, a series of experiments were conducted. The experiments were conducted in three groups with a water depth of 500 mm in the breeding bucket and camera-to-water surface distances of 250, 350, and 450 mm. In each experimental group, the perch were randomly selected for measurement, with ten measurements taken per fish. To ensure the accuracy of the experiment, images of fish with different shapes were obtained for each measurement by adjusting the camera position horizontally on a tripod or by stirring the position of the fish underwater.

Relative error (*RE*) is able to show the relative difference between the estimated and true values:(14)$$RE=\frac{\left|L_{ti}-L_{ai}\right|}{L_{ai}}\times 100\%$$
where $L_{ti}$ represents the distance measured by the algorithm, and $L_{ai}$ represents the actual distance measured manually.

The mean relative percentage error (*MRPE*) was used to evaluate measurement performance.
(15)$$MRPE=\frac{1}{n}\sum_{i=1}^{n}\frac{\left|L_{ti}-L_{ai}\right|}{L_{ai}}\times 100\%$$

Table 1 displays the *MRPE* in the three sets of experiments in comparison to the manual measurement (ML) while considering (CR) and ignoring (IR) the refraction of water. After considering water refraction, the *MRPE* of fish length in Groups 1, 2, and 3 was reduced by 0.21, 0.37, and 0.52%, respectively. As the distance from the camera to the water surface increases, the error obtained from the measurements increases. Consequently, the distance from the camera to the water surface should be considered during measurement.

Figure 14 shows the RE values for the three sets of estimated medium perch lengths, where the top and bottom lines of each box plot indicate the maximum and minimum values, respectively, and the center line of the rectangle connects the *MRPE* values. The *MRPE* of the measurement for each group was within 2%, and the error of any single measurement did not exceed 4%.

## 4. Discussion

This study presents a novel application of binocular vision in the field of underwater fish-length measurement. The research introduces a non-contact method for accurately measuring the length of fish in underwater environments. The experimental results demonstrate that the method has achieved positive outcomes. This section discusses the convenience and effectiveness of the method.

### 4.1. Convenience of the Method

As shown in Table 1, the accuracy improves significantly after considering the refraction of the water. It is worth discussing the calibration of the camera, which is a necessary step before measurement to obtain the camera parameters. The relationship between the world and pixel coordinate systems is established as follows:$$Z_{c}\left[\begin{array}{c} u\\ v\\ 1 \end{array}\right]=\left[\begin{array}{cccc} fx & 0 & u_{0} & 0\\ 0 & fy & v_{0} & 0\\ 0 & 0 & 1 & 0 \end{array}\right]\left[\begin{array}{cc} R & T\\ \vec{0} & 1 \end{array}\right]\left[\begin{array}{c} X_{w}\\ Y_{w}\\ Z_{w}\\ 1 \end{array}\right]=MN\left[\begin{array}{c} X_{w}\\ Y_{w}\\ Z_{w}\\ 1 \end{array}\right]$$
where the matrices *M* and *N* are the camera internal and external references, respectively.

Two calibration methods were used in this study. The first involves calibrating in air, analyzing the effect of water refraction on camera measurements, and restoring the target measurement point to be consistent with that in air. Another method involves placing a calibration plate in water to determine the internal camera parameters by establishing a relationship between the world and pixel coordinate systems. Because this method no longer takes into account the refraction of water, it has the drawback of larger distance measurement errors when using a camera. A recalibration of camera parameters is necessary when there are changes in the water depth or camera height; however, this process may not be convenient. The experiments in this study also proved the feasibility of existing methods.

### 4.2. Effectiveness of the Method

To determine the length of the fish, the first step is to segment the fish from the captured images. In images with multiple fish, each fish exhibits distinct color characteristics, making it challenging for binary segmentation methods [9] to accurately segment all the fish. Section 2.2 explains the selective segmentation of targets with Grab Cut algorithm after changing the contrast of the images. The contour finding algorithm and the minimum external rectangle algorithm can accurately extract the best measurement points: the fish head and tail. Compared with the method where the operator needs to manually mark the fish’s nose and tail fork to estimate the length of the fish [17], it is a more accurate and convenient approach. Many deep learning-based methods [36,37] are applied to fish segmentation, which requires a large dataset. In the measurement, some curved fish will make the error increase [38]. We have improved the measurement accuracy by using the Zhang-Suen refinement algorithm, which can handle fish body bending.

The application of this paper is characterized by its low cost, as it utilizes a single integrated camera for its operation. Savinov [21] used two cameras—one over the aquarium and one on the long side. However, uneven aquarium backgrounds or sharp contrasts can distort the boundaries detected by the fish. To determine the location of the fish, the back wall of the aquarium was painted green [23]. Furthermore, some methods [21,22,23] of measuring the length of fish in aquariums are not applicable in aquaculture engineering. The camera was submerged in water, which resulted in significant distortions [33] in the captured images. Meanwhile, a separate waterproof case needs to be made for the camera. The bending and overlapping of fish in the water also affects the measurement accuracy [18,38]. In future research, we plan to apply the methods described in this paper to the context of factory aquaculture.

## 5. Conclusions

In this paper, an intelligent method for fish-length and -size estimation based on binocular stereo vision was proposed. This method provides a feasible solution to the problem of non-contact fish body-length measurement in the field of intelligent aquaculture. This study utilized the RealSense D435 binocular camera for data acquisition and successfully achieved length measurement. This approach offers a simpler and more convenient installation and usage compared to the use of multiple cameras with varying angles and directions. Subsequently, the problem of inaccuracy in image segmentation was analyzed, and accurate segmentation results were obtained by adjusting the contrast of the image first and performing segmentation later. Two measurement methods for fish in straight and curved states were presented. The effect of water refraction on camera measurement was also considered, resulting in more accurate measurements. Software was developed to measure and calculate the body length of perch. The measurement results showed that the *MRPE* was 0.91%, while the maximum relative error was below 4%. The proposed approach provides a feasible method for the non-contact measurement of underwater fish size, which can facilitate the development and application of machine vision in the field of intelligent aquaculture.

In future work, we will investigate the relationship between the length and mass of perch, which can be obtained directly using the algorithm proposed in this paper.

## Figures and Tables

**Figure 1 sensors-23-06325-f001:**
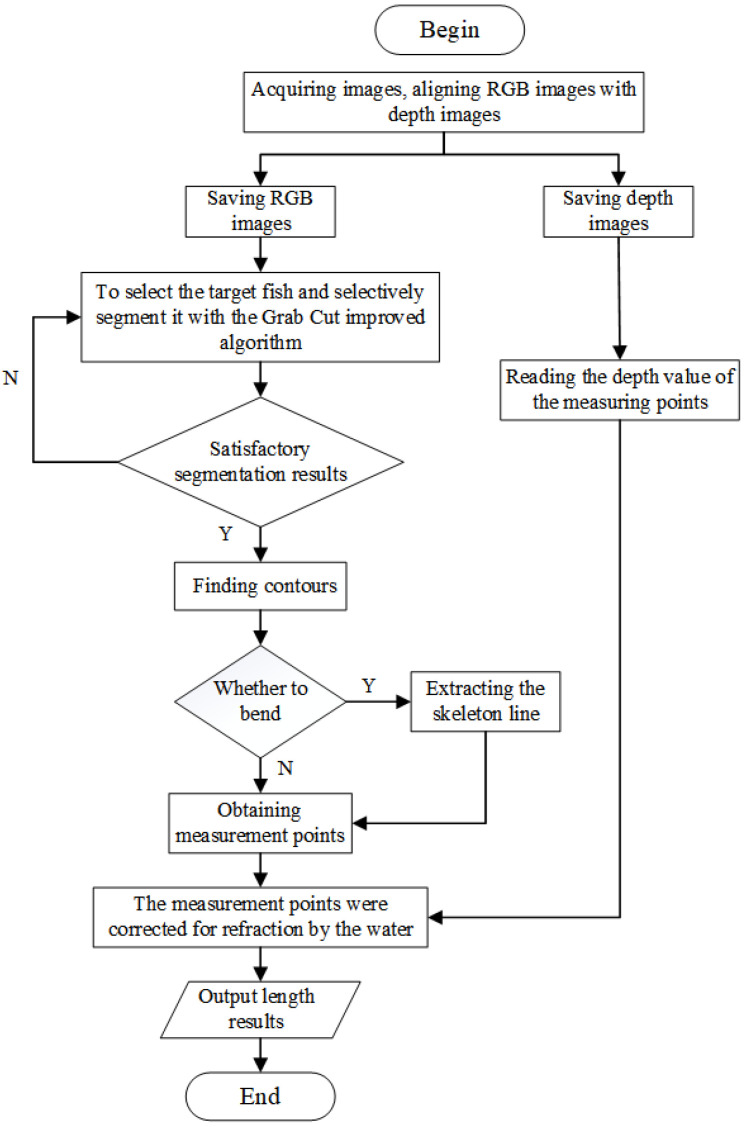
Flowchart of underwater fish body-length measurement method.

**Figure 2 sensors-23-06325-f002:**
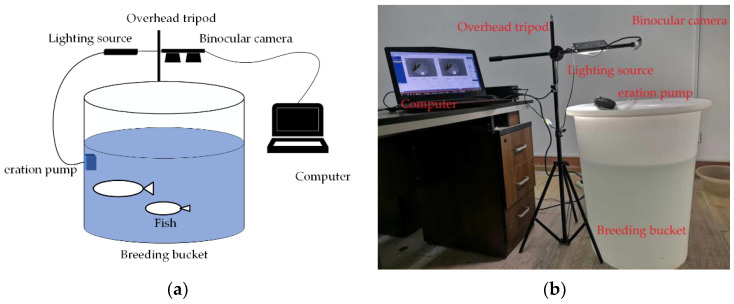
Experimental measurement platform, (**a**) Schematic diagram, (**b**) Image of the actual device.

**Figure 3 sensors-23-06325-f003:**
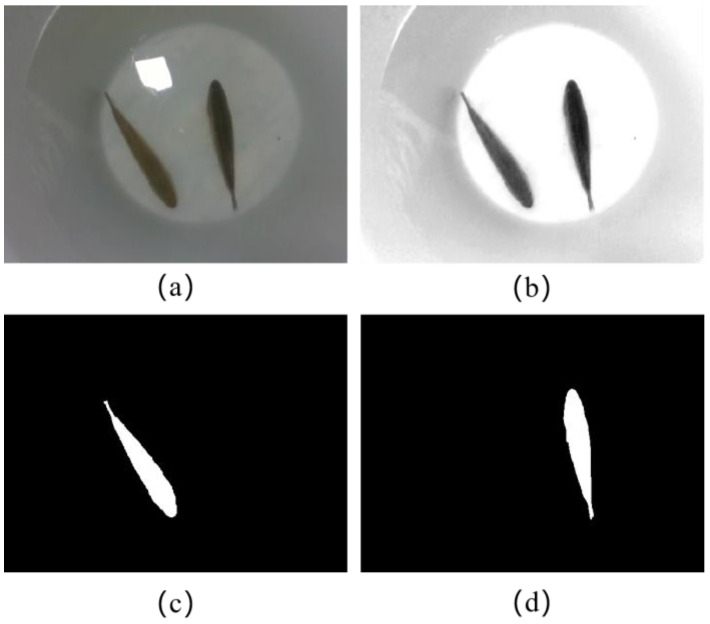
Experimental segmentation results, (**a**) the RGB original image, (**b**) contrast-altered images, (**c**,**d**) segmentation results.

**Figure 4 sensors-23-06325-f004:**
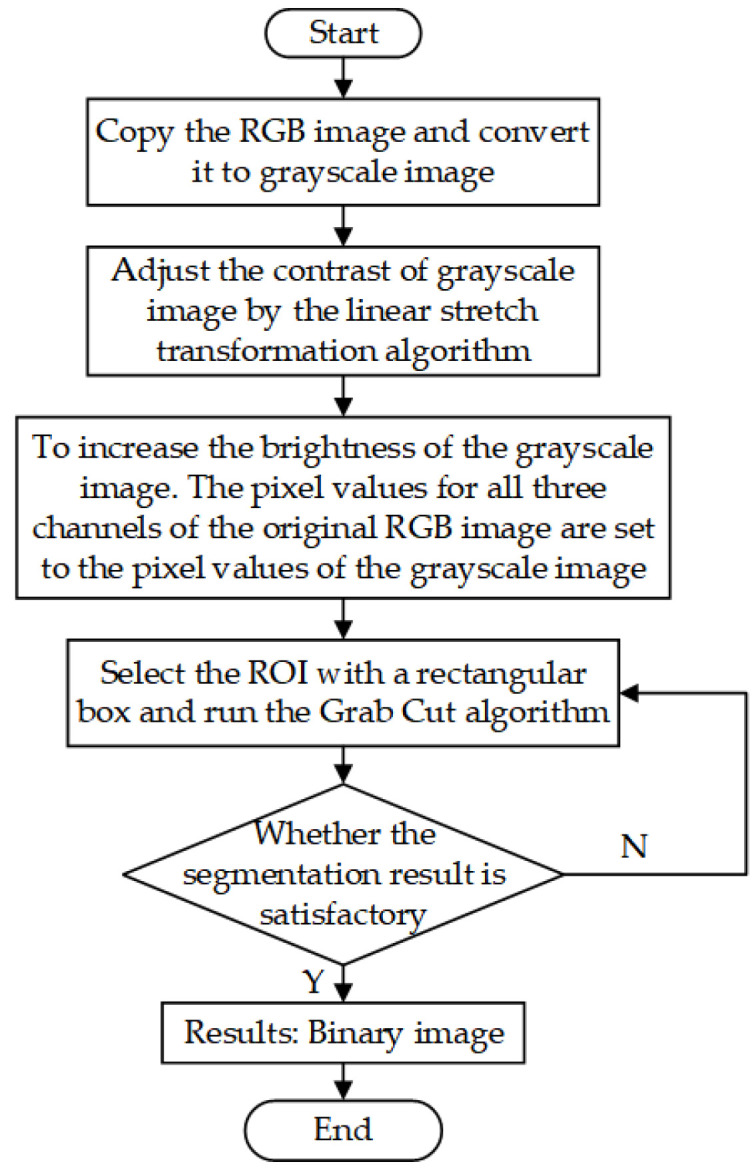
Flowchart of image segmentation algorithm.

**Figure 5 sensors-23-06325-f005:**
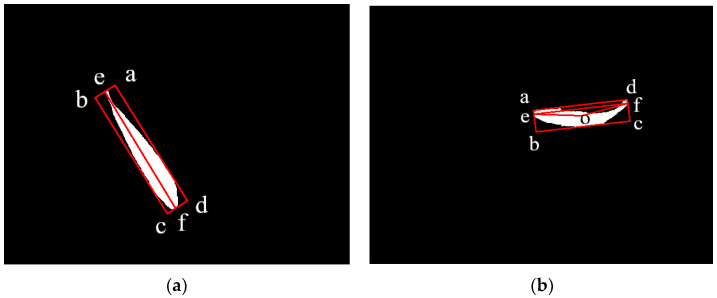
Fish status test results, (**a**) Straight fish, (**b**) Bent fish.

**Figure 6 sensors-23-06325-f006:**
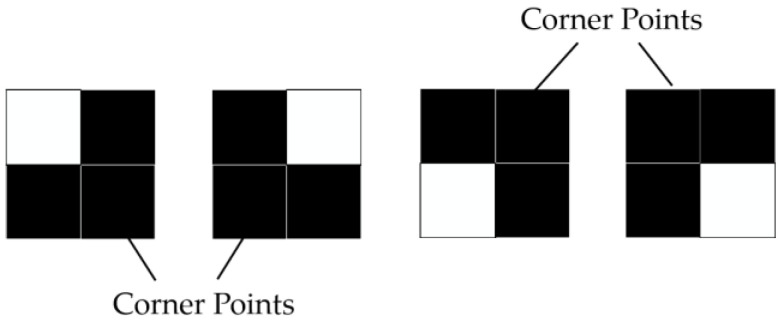
Corner points.

**Figure 7 sensors-23-06325-f007:**
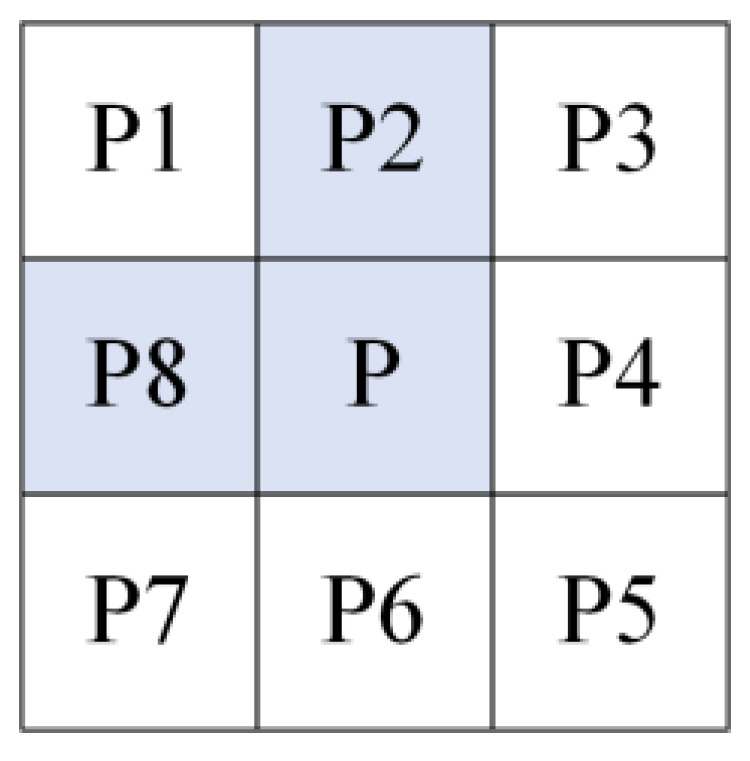
Removal of corner points.

**Figure 8 sensors-23-06325-f008:**
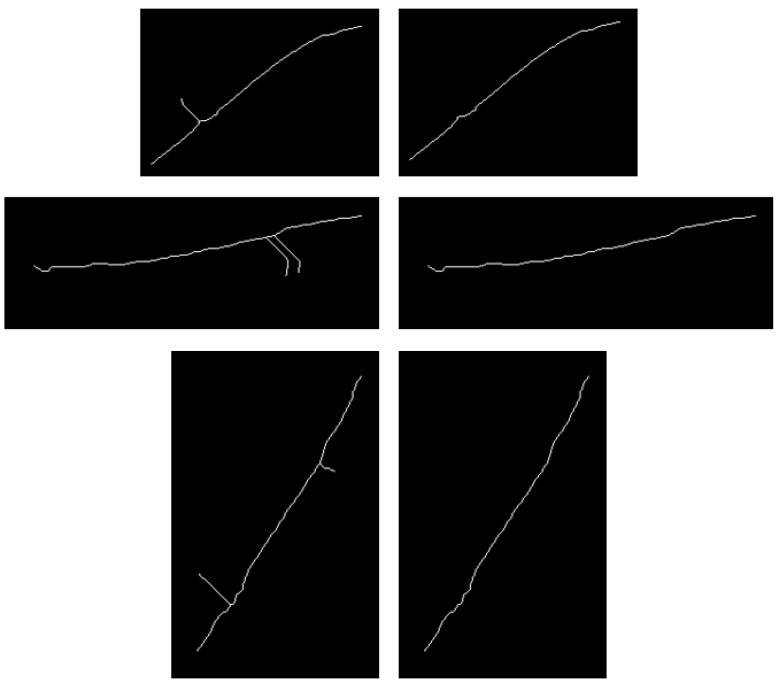
Burr removal results.

**Figure 9 sensors-23-06325-f009:**
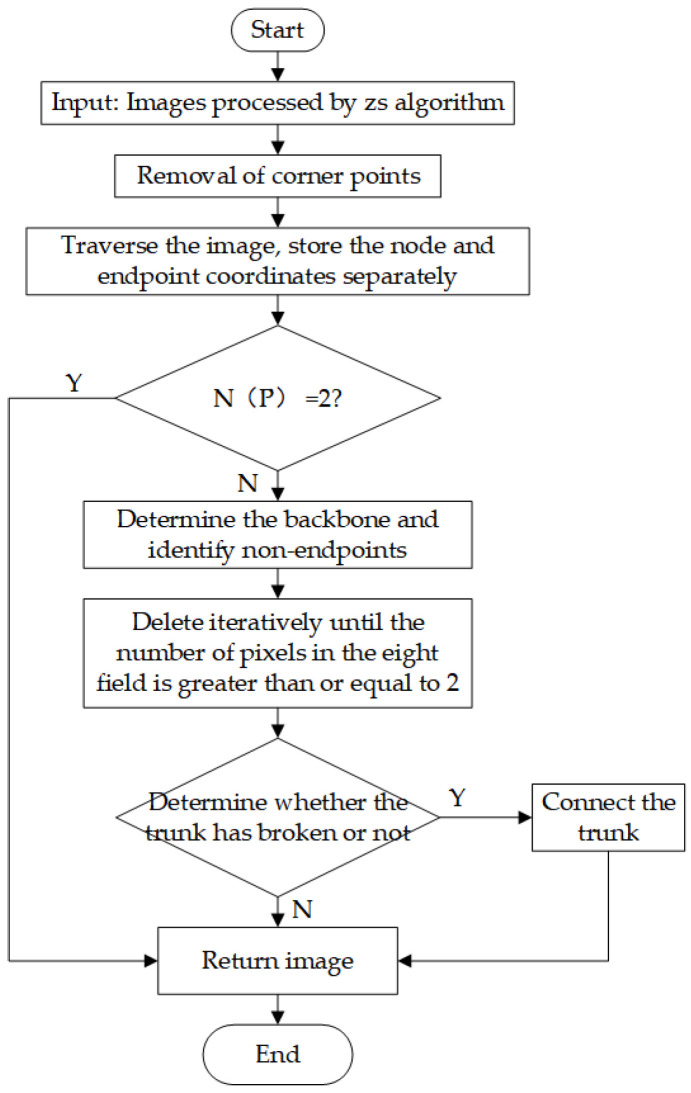
Flowchart of deburring.

**Figure 10 sensors-23-06325-f010:**
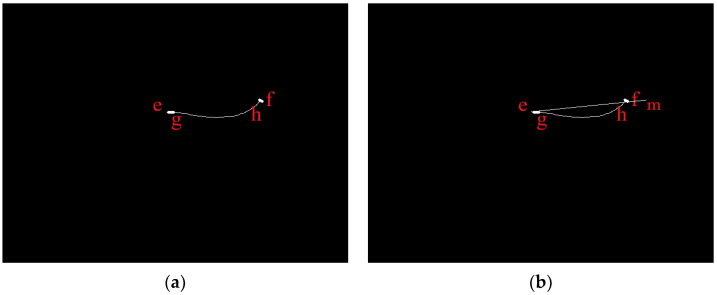
Bending process, (**a**) Processed by Zhang-Suen algorithm, (**b**) Straightening the fish.

**Figure 11 sensors-23-06325-f011:**
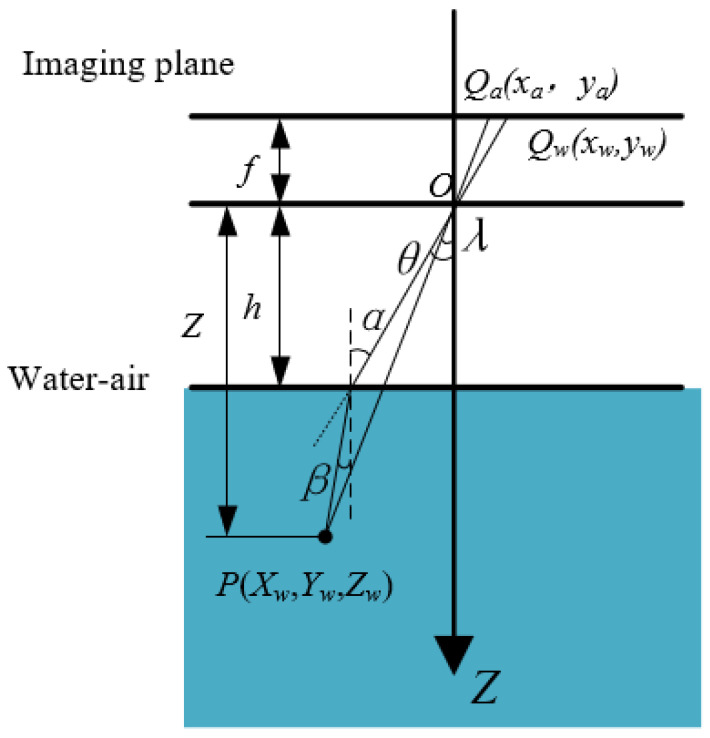
Analysis of light refraction in imaging through water.

**Figure 12 sensors-23-06325-f012:**
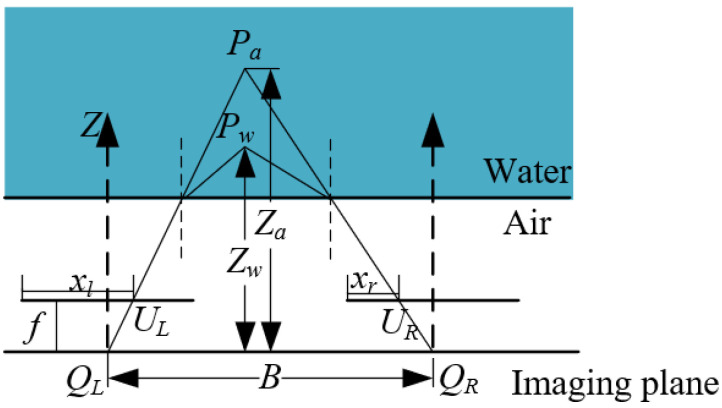
Principle of binocular vision measurement system.

**Figure 13 sensors-23-06325-f013:**
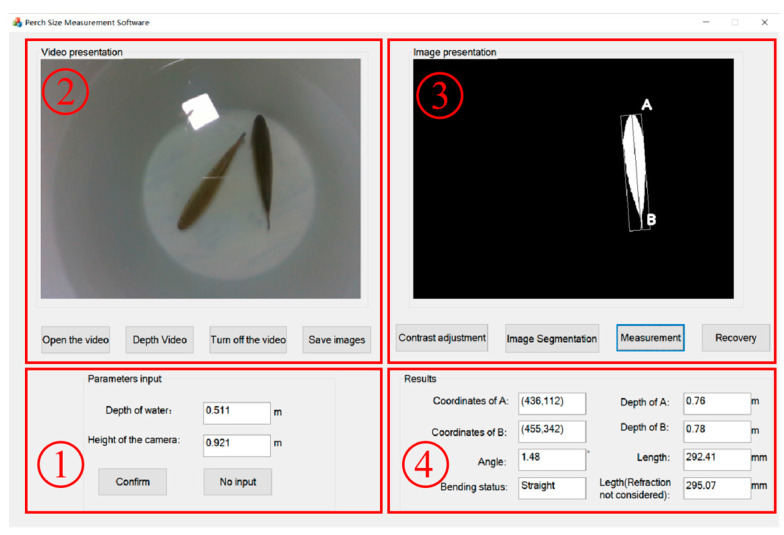
Perch-length measurement software. (**1**) Data input, (**2**) Video presentation, (**3**) Image processing, (**4**) Result display.

**Figure 14 sensors-23-06325-f014:**
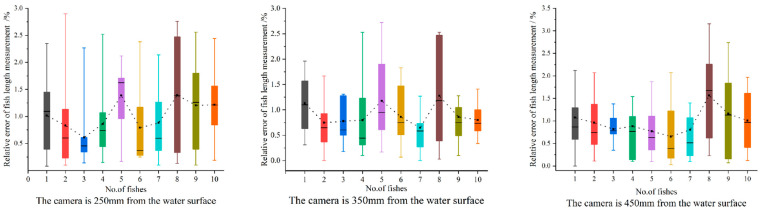
Relative error of fish body-length measurement.

**Table 1 sensors-23-06325-t001:** Relative error of experimental measurements.

No.	ML/mm	Group 1 (250 mm)		Group 2 (350 mm)		Group 3 (450 mm)	
		*MRPE* (CR)	*MRPE* (IR)	*MRPE* (CR)	*MRPE* (IR)	*MRPE* (CR)	*MRPE* (IR)
1	255	1.02%	1.17%	1.13%	2.07%	1.07%	1.68%
2	300	0.83%	1%	0.65%	0.8%	0.8%	0.99%
3	295	0.61%	0.77%	0.86%	0.96%	0.65%	0.83%
4	270	0.86%	0.92%	0.74%	1.44%	0.96%	1.92%
5	292	1.38%	1.71%	0.79%	0.97%	0.88%	1.76%
6	282	0.79%	1.06%	0.77%	0.89%	0.82%	0.97%
7	294	0.87%	1.2%	1.12%	1.27%	0.77%	1.36%
8	305	1.27%	1.31%	0.82%	1.03%	0.82%	1.3%
9	320	1.21%	1.57%	0.76%	1.63%	1%	1.76%
10	304	0.9%	1.13%	0.96%	1.27%	1.33%	1.78%
total		0.97%	1.18%	0.86%	1.23%	0.91%	1.43%

## Data Availability

The data presented in this study are available on request from the corresponding author.

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
