# Peer review of "In-Water Fish Body-Length Measurement System Based on Stereo Vision"

_sensors, 2023, doi:10.3390/s23146325_

Round 1

Reviewer 1 Report

In general, the paper is well written and uses a good English. I think it contains all relevant technical details. It was written excellently what was done. The reviewer would like to read something about the applied programming language (MATLAB, Python, etc.) or libraries. Moreover, a computational analysis of the proposed method would be welcomed. I have two questions. 1.: How does the intensity of the lighting source influence the performance of your method? Can lighting sources applied in a real-life system? 2.: Let's say that one fish is at 200 mm and another fish is at 400 mm. Does the performance depend on the depth where the fish is? What is the maximal depth which can be achieved? 

Reviewer 2 Report

1.  The authors propose a non-contact measurement method that utilizes binocular stereo vision to accurately acquire the body length of underwater fish. Binocular cameras capture the RGB and depth images to acquire RGB-D data on the fish.

2. In the figure 2, experimental measurement platform should be demonstrated in detail.

3.   In the figure 4, flow chart of image segmentation algorithm should be demonstrated in detail.

4.      In the figure 5, bend test results should be demonstrated in detail.

5.      In the figure 6, corner Points should be demonstrated in detail.

6.     The authors are suggested to highlight the contributions of the proposed work, compared to the prior works. A detailed discussion about prior works are suggested to add.

7.  The manuscript has 14 pages; the number of the pages should be increased.

8.      Revise the English thoroughly before submission.

Moderate editing of English language required.

Round 2

Reviewer 2 Report

no further comment.